# Management of Ground-Glass Nodules: When and How to Operate?

**DOI:** 10.3390/cancers14030715

**Published:** 2022-01-29

**Authors:** Young Tae Kim

**Affiliations:** 1Department of Thoracic and Cardiovascular Surgery, Seoul National University Hospital, Seoul 03080, Korea; ytkim@snu.ac.kr; Tel.: +82-2-2072-3161; 2Cancer Research Institute, Genomic Medicine Institute, Medical Research Center, Seoul National University College of Medicine, Seoul 03080, Korea

**Keywords:** GGN, surgery, lung cancer

## Abstract

**Simple Summary:**

An increasing number of lung cancer screening programs have detected the frequent occurrence of small pulmonary ground-glass nodules (GGNs). If GGN is an incidental finding, it should be followed according to the guidelines. A multidisciplinary team discussion should be initiated if a new solid component develops or the solid portion grows on follow-up CT. Preoperative attempts to biopsy solid components in part-solid GGNs are often not feasible and not helpful. If malignancy is suspected, a surgical biopsy with the guidance of various localization methods is recommended. Once the GGN is confirmed to be malignant, sub-lobar resection may be reasonable in the majority of cases, and the extent of lung resection should be determined based on the CT finding or intraoperative frozen section examination using special inflation technique. Although rare, the recurrence in the remaining lobe can occur especially in patients with high risk histologic features, which currently cannot accurately diagnosed either pre- or intra-operatively.

**Abstract:**

With the increased popularity of low-dose computed tomography (LDCT), many patients present with pulmonary ground-glass nodules (GGNs), and the appropriate diagnostic and management strategy of those lesions make physicians be on the horn of the clinical dilemma. As there is not enough data available to set universally acceptable guidelines, the management of GGNs may be different. If GGN is an incidental finding through LDCT, the lesion should be followed according to the current guidelines. We recommend a multidisciplinary team discussion to be initiated if a new solid component develops or the solid portion size grows on follow-up CT as the risk of malignancy is high. Attempts to preoperatively biopsy solid components in part-solid GGNs are often not feasible and not helpful in clinical settings. Currently, if malignancy is suspected, a surgical biopsy with the guidance of various localization methods is recommended. If malignancy is confirmed, sub-lobar resection may provide an excellent oncologic outcome.

## 1. Introduction

Since the introduction of low-dose computed tomography (LDCT), an increasing number of lung cancer screening programs have detected the frequent occurrence of small pulmonary ground-glass nodules (GGNs). In Asia, LDCT is commonly used for screening purposes, even for the non-smoking population, among whom GGNs are often found [1,2]. However, the prevalence of GGN in the other part of the world was reported to be low, but most data were driven from randomized controlled lung cancer screening studies, where most of the participants were smokers [3,4,5]. Although GGNs are non-specific radiologic findings which can be manifestations of benign conditions such as focal interstitial fibrosis, eosinophilic pneumonia, aspergillosis, bronchiolitis obliterans organizing pneumonia, or Wegener granulomatosis, some GGNs are adenocarcinomas or its precursors, atypical adenomatous hyperplasia (AAH) [6]. Consequently, clinicians often face a clinical dilemma. This article will review and analyze current evidence and offer evidence-based recommendations on the management of GGNs focusing on the Asian population.

## 2. Predicting Malignant GGN

Pulmonary adenocarcinomas can manifest as solid, pure, or part-solid GGNs. Among them, persistent part-solid GGNs have been shown to have the highest probability of being pulmonary adenocarcinomas [7]. The biological nature of lung adenocarcinoma manifested as a GGN is known to be indolent, which has a greater survival rate as compared to solid type lung adenocarcinoma. The size of the solid portion rather than the size of the whole GGN determines the prognosis in the part-solid GGN [8]. Accordingly, in the 8th edition of the IASLC staging system, the T stage of lung cancer presenting as part-solid GGN is determined based on the size of the invasive portion instead of the size of the entire GGN [9]. This new T staging system was validated by several studies [10,11]. However, recently published data using Japanese Clinical Oncology Group (JCOG) 0201 datasets found a favorable prognostic impact of the GGO component, regardless of the solid component size in tumors, and as a consequence, the T classification may need further refinement in the future [12].

## 3. The First Dilemma: When to Intervene?

If a GGN is an incidental finding on a chest CT, the follow-up based on several guidelines is recommended [13]. If the GGN disappears or decreases in size, the lesion is presumed to be benign. However, if the GGN remains persistent for several months, a neoplastic condition should be suspected (Figure 1). There is no established consensus as to when an invasive investigation of a GGN should be initiated.

If the size of the GGN is small, determining the optimal management approach becomes even more arguable. While some surgeons prefer resecting even a small, pure GGN [14], others believe that a GGN should only be removed if there is evidence of growth. This dilemma stems from the fundamental question: How does cancer become invasive? We recently analyzed whole-genome sequencing data of lung adenocarcinoma and found a long latency period from the time of acquisition of driver oncogene mutation to the date of clinical diagnosis of lung cancer. It suggests that after cells acquire driver mutations, they stay quiescent for a long time before they obtain secondary mutations, through which the tumor evolves to clinically invasive lung cancer [15]. If a pure GGN represents such a quiescent period of cancer, at what point does a GGN become invasive? How can we detect that moment?

Can we use FDG PET-CT to determine the invasiveness of the GGN? As the SUVmax of most GGNs is usually very low, especially if the size of the solid component is small, the utility of FDG PET-CT in discriminating benign vs. malignancy is limited. Furthermore, preoperative FDG PET-CT seems to have limited utility in detecting lymph node and distant metastases in patients with part-solid GGNs with a solid portion size of 3 cm or smaller [16]. Therefore, it is our current practice not to perform preoperative FDG PET-CT for patients with GGNs to either assess the nature of GGN or evaluate the presence of metastasis in the lymph node or distant organs.

A Japanese study has suggested that if large peripheral GGNs continue to appear during follow-up screenings, those GGNs tend to be lung cancer [17]. Our group previously published a retrospective study where we observed GGNs detected on CT scans of patients who had previously undergone major lung resection. As the patients had previously undergone lung surgery, the threshold for deciding on another round of surgery was high for both patients and doctors, and thus provided an extended period of time to observe the GGNs. We found that the presence of a solid component to the GGN was the only factor that could predict growth, and the majority of GGNs that showed evidence of growth were adenocarcinoma [18]. 

The solid portion of the GGN matters: if a new solid component develops or the solid portion of the GGN shows evidence of growth on follow-up, the probability of the malignant lesion becomes high. Therefore, it is our practice that if a solid portion of a GGN becomes prominent or persistent, we initiate a multidisciplinary team discussion with the patient and the family to determine a treatment approach [19]. Such a conservative approach does not seem to increase mortality while decreasing the risk of performing unnecessary surgery for screen-detected GGNs [4].

## 4. The Second Dilemma: Can We Make a Preoperative Tissue Diagnosis?

Ideally, preoperatively acquiring a correct tissue diagnosis allows us to avoid unnecessary surgery if the GGN is benign and helps us determine the extent of surgery necessary if the GGN is malignant. The second dilemma begins here: Is it possible to make a preoperative tissue diagnosis?

It is suggested that preoperative diagnosis of GGNs by trans-thoracic fine-needle aspiration biopsy can be performed with reasonable diagnostic accuracy [20,21]. However, both trans-bronchial and trans-thoracic needle biopsy methods have limitations for the pathologic confirmation of GGNs as they often miss small invasive foci [22,23]. Shimizu and colleagues conducted a retrospective study where the value of preoperative percutaneous CT-guided fine-needle aspiration biopsy (CTNB) was evaluated in patients with peripheral lung cancers smaller than 2 cm presented as GGN. Among the ground-glass opacity (GGO)-dominant group, the diagnostic yields were as low as 35.2% for lesions smaller than 10 mm, 50.0% for those 11–15 mm, and 80.0% for those 16–20 mm [24]. Although the authors concluded that CTNB is a useful diagnostic tool for peripheral small lung cancers with GGO, the poor preoperative diagnostic yield in GGO-dominant lung cancer seems to be suboptimal for clinical use. In the other retrospective study from Japan, the authors tried to find the diagnostic yield of CTNB in pure GGN. Although the calculated diagnostic accuracy was 95% (63/66), it is noteworthy that seven out of eight lesions, where the needle biopsy results were AAH, were actually adenocarcinoma. Moreover, of the 15 lesions with a benign diagnosis using CTNB, as many as nine patients underwent surgery or RT, three on follow-up, due to a suspicion of cancer. Two patients were missed during follow-up, due to an erroneous belief that they did not have cancer based on a benign needle biopsy result [25]. This real-world situation occurs in clinical practice. Although needle biopsy results may indicate non-malignancy, a negative needle biopsy result should be suspected as a false-negative if a lesion is clinically suspicious for malignancy.

In my institution, we retrospectively evaluated the diagnostic accuracy and complications of CTNB for 172 small lung nodules. With CTNB, 160 nodules were correctly diagnosed, and three were false negatives. Diagnostic accuracy, sensitivity, and specificity were 98.2%, 96.8%, and 100%, respectively. However, among the entire cases, only 11 patients had GGNs. The study population was consecutive samples, and therefore it reflected my hospital’s clinical practice where CTNB is not usually recommended for GGNs. Indeed, we found a significantly higher risk of hemoptysis in GGNs (OR = 5.10) [26].

In another study, we evaluated 1116 consecutive lung lesions which had undergone CTNB. Initially, the non-specific benign diagnosis was made in 226 lesions, and of those, 24 (10.6%) were finally confirmed as malignancies (negative prediction value (NPV) = 89.4%). Through multivariate analysis, a part-solid GGN (OR = 3.95) was one of the factors which were significantly associated with a false-negative result [27]. Such observations suggest that needle biopsy should not be relied on for part-solid GGN as a result has very low NPV and can result in a high number of false-negative results, leading to erroneous treatment decisions.

Ikezawa and colleagues have recently shown that endobronchial ultrasonography images and virtual bronchoscopy could be obtained for 156 (92%) of 169 GGO predominant-type lesions, with 116 (69%) being successfully diagnosed using this method [28]. Unfortunately, however, there is not enough data to support this practice.

Accordingly, preoperative biopsy for GGNs is not used in our current practice, and if the suspicion for malignancy is high on CT, a surgical biopsy is performed without a preoperative needle biopsy. We retrospectively reviewed 85 PSNs with solid components >5 mm on CT who underwent surgery. Preoperative CTNBs were performed for 41 PSNs (biopsy group), and CT assessment-based direct resections were performed for 44 PSNs (direct surgery group). In the biopsy group, the overall sensitivity, specificity, and accuracy for the diagnosis of adenocarcinoma were 78.9% (30/38), 100% (1/1), and 79.5% (31/39), respectively. In the direct surgery group, the respective values for the diagnosis of adenocarcinoma were 100% (38/38), 0% (0/6), and 86.4% (38/44), respectively, and there were no significant differences in diagnostic accuracy (*p* = 0.559) between the two groups [29]. Given such results, our current practice is to use CT assessment-based direct surgical resection for PSNs with solid part >5 mm.

For those lesions with pure GGNs, we do not resect immediately and recommend regular follow-up. However, the decision is difficult to make for persistent part-solid GGNs with solid components ≤5 mm. When we retrospectively analyzed 125 surgically resected persistent PSNs with solid components ≤5 mm, we found that surgery was performed after observing interval growth in 54 patients, whereas 71 patients underwent surgery immediately. In the interval growth group, 30 patients showed increased whole GGN size, 10 increased solid parts, and the remaining 14 showed growth in both portions. We found no significant differences between these two groups regarding recurrence-free survival (*p* = 0.451) and overall survival (*p* = 0.185) [30]. Based on this result, we learned that the policy following GGN through interval growth and then resecting does not negatively influence the prognosis of patients with persistent part-solid GGNs with solid components ≤5 mm. However, if the patient becomes anxious, we tend to operate earlier.

## 5. The Third Dilemma: How to Localize GGN?

During a surgical biopsy, especially in minimally invasive surgery, GGNs are often difficult to palpate. Thus, preoperative marking techniques have been utilized to localize the GGNs [31]. It can be done either percutaneously or trans-bronchially, using various materials including a dye, colored collagen, barium, lipiodol, micro coil, metallic wire, or fiducial [32]. In our center, we have used various methods for preoperative localization with reasonable accuracy. For the deep-seated GGN, we used a preoperative CT-guided lipiodol marking near the site from the GGN and used intraoperative fluoroscopy to assist determining sufficient resection margin from the GGN. However, in the centers without hybrid operation theater, such technique is cumbersome as the patients require additional travel to CT room. Therefore, currently, we favor using the electromagnetic navigational bronchoscopy (ENB) guided dye marking technique. In our series, 29 ENB-guided dye markings were done for 24 nodules in 20 patients, and the success rate of nodule localization was 95.8% (23/24). All marked nodules were completely resected thoracoscopically. Technically, we found a pathway with an obtuse approach angle should be selected to increase the navigation accuracy [33]. If the lesion is located in an area that requires an acute angle, the localization is not feasible, and we often have to use CT guided technique.

## 6. The Fourth Dilemma: How Much Should We Resect?

Determining the extent of surgical resection is the next decision point. Since the previous randomized study comparing lobectomy versus limited resection conducted by the Lung Cancer Study Group, lobectomy has been the treatment of choice for NSCLC [34]. However, the study is more than 30 years old and was conducted at a time without PET, EBUS, or high-resolution CT scans. Obviously, at that time, there was no knowledge of GGNs.

There is some evidence demonstrating the inferior outcome of sublobar resection. However, several studies have shown that limited resection may be beneficial, especially in early-stage lung adenocarcinoma, including GGN [35]. Nakao and colleagues, however, have reported the development of adenocarcinomas in the surrounding area of the initial resection site after more than five years in four out of 26 patients [36]. Nevertheless, as GGNs usually show a favorable prognosis, limited resection may provide an excellent oncologic outcome in appropriately selected patients [37]. JCOG recommended lobectomy for GGO larger than 3 cm, segmentectomy for lesions between 2 and 3 cm, and wedge resection for GGO smaller than 2 cm if a C/T ratio is less than 0.5. If a C/T ratio is 0.5–1.0, segmentectomy is recommended even in a GGO less than 2 cm. Several prospective randomized trial studies are ongoing in Japan, intending to find a definitive answer (JCOG0804, JCOG1211, JCOG0802).

In our practice, we favor treating minimally invasive adenocarcinoma (MIA) or earlier lesions with wedge resection. For invasive adenocarcinoma, we prefer to perform anatomic resection (segmentectomy or lobectomy). If a patient has GGN lesions, and if the lesion is suspicious for lung cancer, we recommend an excisional biopsy, which is usually performed through a wedge resection. If cancer is found through examining a frozen section, we have to decide whether or not to proceed with anatomical resection. As mentioned above, if the lesion is MIA or an earlier lesion, we do not proceed further and finish the operation. However, if it is invasive adenocarcinoma, we perform either segmentectomy or lobectomy.

Necessary for this process is a method to distinguish whether the lesion is MIA or invasive adenocarcinoma. Several studies have suggested that the area of radiologic ground-glass components of part-solid GGNs overlap onto the lepidic, pre-invasive components of pulmonary adenocarcinomas in a pathologic slide, while solid components frequently represent the invasive area. However, CT findings often do not correlate with pathology [38]. Intraoperative frozen section diagnosis is an alternative method, but the problem is that deflated lung specimens can make accurate diagnosis difficult [39]. A technique of inflating the lung specimen with the embedding medium has been used, which allows for better interpretation and facilitates correct diagnosis in the frozen section [40]. Our center has been using this method and has found a high diagnostic accuracy with a concordance rate of 90.6% between frozen and permanent pathology (Figure 2). Based on our experience, our current practice is to perform a wide wedge resection of the GGNs and send the specimen for the frozen section using the embedding medium inflation method. If the result of the frozen examination is pre-invasive lesions (benign, AAH, AIS, or MIA), we do not perform additional resection. If invasive adenocarcinoma is diagnosed, we prefer to proceed with anatomic lung resection with systematic lymph node dissection.

Additionally, in cases of deeply located GGNs, where wedge resection is not technically feasible, direct segmentectomy without wedge biopsy is recommended for diagnosis and treatment. Various techniques can be used to guide a correct resection margin. Previously, we used preoperative CT guided lipiodol marking, and intraoperative fluoroscopy was used to guarantee a sufficient resection margin from the tumor. Although the approach has worked well, it is cumbersome as patients must undergo a CT scan before coming to the OR, and we are currently using a 3D-reconstruction program to explore the distance from the lesion to the intersegmental plane. We prefer to have a sufficient parenchymal resection margin beyond the intersegmental plane, and a precise delineation of the intersegmental plane is obtained by using indocyanine green (ICG) injection during the operation.

## 7. The Fifth Dilemma: Concerns of Sublobar Resection

Although the sublobar resection may be appropriate for the treatment of GGN in oncologic aspects, there are several clinical issues to be addressed.

In a recent study, the authors investigated 1497 patients who underwent either lobectomy or sublobar resection for T1N0M0 lung ADC. After propensity score matching, they found sublobar resection was associated with a significantly higher recurrence in patients with STAS, whereas such phenomenon was not observed in those without STAS [41]. They also investigated the effect of margin-to-tumor ratio on recurrence patterns. Among patients with STAS-negative tumors, if the margin-to-tumor ratio was one or higher, recurrence was rare, and no locoregional recurrence was observed. On the contrary, the risk of recurrence among patients with STAS-positive tumors was high regardless of the margin-to-tumor ratio. Such observation suggests that segmentectomy is inappropriate in STAS-positive lung cancers even though the resection margin is sufficiently long enough. The dilemma is that one cannot accurately predict the presence of STAS preoperatively or intraoperatively. Although the authors suggested the sensitivity and specificity for detecting STAS during the frozen section were 71% and 92%, the inter-rater reliability was only 67%.

Other poor prognostic factors of recurrence in the remaining segments, such as the presence of visceral pleural invasion, micropapillary, solid, or mucinous histology, are difficult to be determined at the time of frozen examination. It is noteworthy knowing subtypes of adenocarcinoma at the time of frozen section may also be helpful to predict occult lymph node metastasis. In the study from MSKCC, the authors discovered that a significant number of patients diagnosed as cN0 stage actually had occult lymph node metastasis. They found that micropapillary histology and solid histology are highly related to the presence of LN metastasis [42].

Therefore, we must realize that the risk of recurrence in the remaining segments after segmentectomy exists in patients with poor prognostic factors, which currently cannot be diagnosed before or during the operation.

## 8. Conclusions

To summarize, if GGN is an incidental finding through LDCT, the lesion should be followed according to the current guidelines. If a new solid component develops or the solid portion size grows on follow-up CT, the risk of malignancy is high, so surgical resection should be discussed in the multidisciplinary team. Although several CT findings can differentiate between pre-invasive and invasive lesions, those findings have not yet proven sufficiently reliable to guide the management plan for GGNs. In addition, attempts to preoperatively biopsy solid components in part-solid GGNs are often not feasible and, therefore, not helpful in clinical settings. Currently, the best practice for the management of GGNs is to carefully monitor patients through follow-up CTs, and if malignancy is suspected, to perform a surgical biopsy with the guidance of various localization methods or other innovative methods. Determining the appropriate extent of lung parenchymal resection may require additional randomized study results, but limited resection may provide an excellent oncologic outcome. Even though most GGNs can be treated with sublobar resection, the recurrence in the remaining lobe exists, especially if there are high-risk histologic features, which are currently not accurately diagnosed either pre- or intra-operatively.

## Figures and Tables

**Figure 1 cancers-14-00715-f001:**
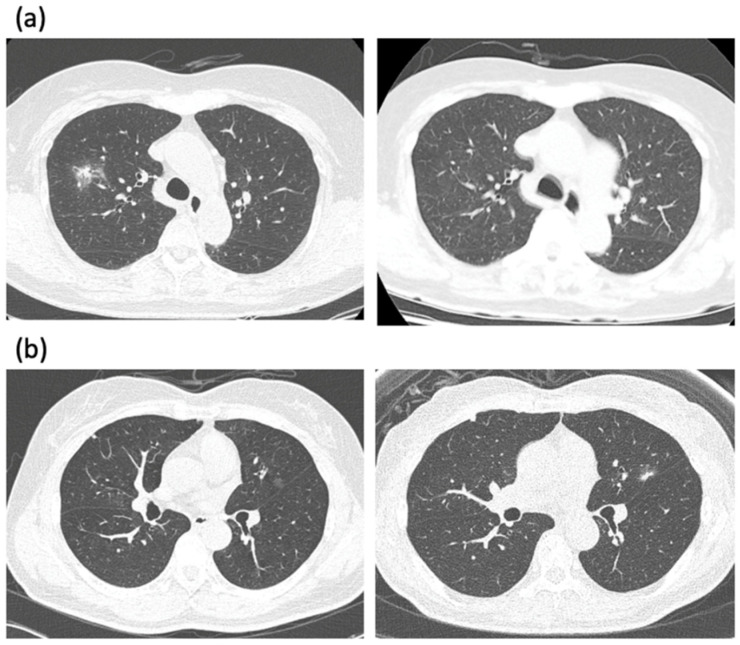
(**a**) A low-dose CT scan of an 80-year-old female presented with incidentally found part-solid GGN in the right upper lobe. The patient refused treatment and decided to observe it. The PSN decreased and almost disappeared three years later. (**b**) A low-dose CT scan of a 67-year-old female presented with incidentally found pure GGN in the left upper lobe. Note the growth of GGN with a newly developed solid portion after 6 years. The lesion was resected with wedge resection, and left upper lingular segmentectomy was subsequently performed. The pathology revealed adenocarcinoma T1a with a total tumor size of 1.5 cm and the invasive portion of 0.6 cm.

**Figure 2 cancers-14-00715-f002:**
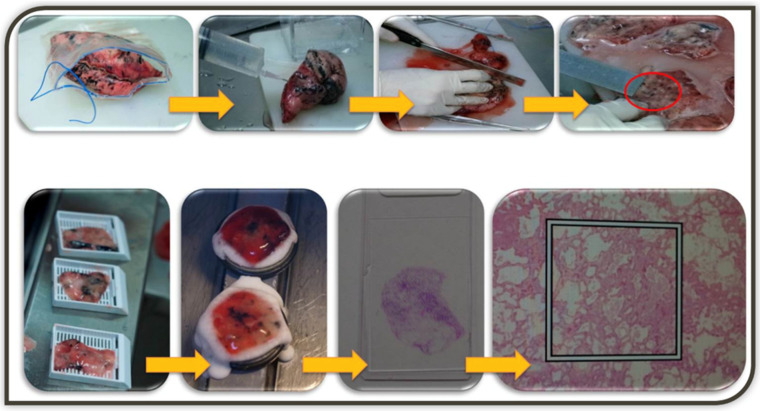
The illustration of embedding medium inflation technique. Lung wedge resection specimen is inflated by injection of 2:3 diluted embedding medium (Tissue-Tek OCT, Sakura Finetek-USA, CA) for cryosection using as 18-gauzed injection needles through the pleura until the lung tissue expands. Serial gross sections are placed, and the slice with GGN is embedded in Cryometirx and frozen at −2 °C. Frozen tissue block is cut into 5 μm thick sections, treated with 95% alcohol, and subsequently stained with hematoxylin and eosin. After frozen section diagnosis, the specimens are immersed in 10% neutral formalin for the permanent paraffin section.

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
