# Peer review of "Management of Ground-Glass Nodules: When and How to Operate?"

_cancers, 2022, doi:10.3390/cancers14030715_

Round 1
Reviewer 1 Report
The author provides a comprehensive review on the management of ground-glass nodules (GGN) focussing on 4 dilemmas: timing of intervention, preoperative tissue diagnosis, localization of GGN and finally, extent of resection. This paper is well written and clinically relevant with updated information on current treatment of ground-glass nodules.
Comments:
- this review focusses on an Asian population; ground-glass nodules are less common in Europe and North America with probably, also a different behavior. As Cancers is an international journal, regional (intercontinental) variation should be addressed
- 4th dilemma: spread through air spaces (STAS) may be present with early lung cancers and is known to be a negative prognostic factor requiring more extensive resection (lobectomy); this should be mentioned and discussed; is frozen section accurate in this setting?
- to make this review complete 2 relevant, recent references should be included and discussed:
1. Sihoe A. Should sublobar resection be offered for screening-detected lung nodules? Transl Lung Cancer Res 2021; 10:2418-26
2. Fu F et al. Computed tomograhy density is not associated with pathological tumor invasion for pure ground-glass nodules. J Thorac Cardiovasc Surg 2021; 162:451-9
Author Response
Thank you very much for reviewing my manuscript and providing valuable comments.
I revised the paper by incorporating issues addressed as follows.
- this review focusses on an Asian population; ground-glass nodules are less common in Europe and North America with probably, also a different behavior. As Cancers is an international journal, regional (intercontinental) variation should be addressed
- Answer) Thank you very much for the comment. I agree with and revised manuscript addressing this issue as follows. "This article will review and analyze current evidence and offer evidence-based recommendations on the management of GGNs focusing on the Asian population.”
- 4th dilemma: spread through air spaces (STAS) may be present with early lung cancers and is known to be a negative prognostic factor requiring more extensive resection (lobectomy); this should be mentioned and discussed; is frozen section accurate in this setting?
- Answer) Thank you for bringing and essential issue. I updated the issue of STAS.
- to make this review complete 2 relevant, recent references should be included and discussed: 1. Sihoe A. Should sublobar resection be offered for screening-detected lung nodules? Transl Lung Cancer Res 2021; 10:2418-26 2. Fu F et al. Computed tomograhy density is not associated with pathological tumor invasion for pure ground-glass nodules. J Thorac Cardiovasc Surg 2021; 162:451-9
- Answer) I added those references.
Reviewer 2 Report
Dear authors,
Thank you for giving me the opportunity to review your article. This review is well written with adequate references. However, I feel a little bit disapointed after reading your article. You missed to explain the main title of your article!
How to follow? After reading your article, I have no idea how I should follow a GGO. How long the follow up, which interval?, signs of benign lesion among a GGO, risk factors, etc. Low-dose CT-scan or injected? Which population?
Which patients should benefit resection (growth after X months, limit of the size of the lesion? Size of the solid components..)
page 4: you should develop a little but more the advantages and inconvenient of your technique to localize the GGO (ENB, ok but why?)
Author Response
Thank you very much for reviewing my manuscript and providing valuable comments.
I revised the paper by incorporating issues addressed as follows.
- Thank you for giving me the opportunity to review your article. This review is well written with adequate references. However, I feel a little bit disapointed after reading your article. You missed to explain the main title of your article!
- How to follow? After reading your article, I have no idea how I should follow a GGO. How long the follow up, which interval?, signs of benign lesion among a GGO, risk factors, etc. Low-dose CT-scan or injected? Which population?
- Answer) Thank you very much for the comment. As a matter of fact, I did not write the follow-up protocol in detail. Indeed, as a thoracic surgeon, I am not in a good position to address the issue. Therefore, I changed the title.
- Which patients should benefit resection (growth after X months, limit of the size of the lesion? Size of the solid components..)
- Answer) As you know, there is no clear cut-off value of those. I hope my article may be helpful for our readers to get a comprehensive understanding of GGN, from which a relevant clinical decision is made.
- page 4: you should develop a little but more the advantages and inconvenient of your technique to localize the GGO (ENB, ok but why?)
- Answer) We prefer ENB as it is more convenient in our hands. Certainly, ENB has disadvantages, and I added a couple of weaknesses to the manuscript.
Round 2
Reviewer 2 Report
Ok for me